

# A time-dependent momentum-resolved scattering approach to core-level spectroscopies

Krissia Zawadzki[1], Alberto Nocera[2] and Adrian E. Feiguin[3]⋆

**1** School of Physics, Trinity College Dublin, College Green, Dublin 2, Ireland
**2** Stewart Blusson Quantum Matter Institute, University of British Columbia,
Vancouver, British Columbia V6T 1Z4, Canada
**3** Department of Physics, Northeastern University, Boston, Massachusetts 02115, USA

⋆ a.feiguin@northeastern.edu

## Abstract

While new light sources allow for unprecedented resolution in experiments with X-rays, a theoretical understanding of the scattering cross-section is lacking. In the particular case of strongly correlated electron systems, numerical techniques are quite limited, since conventional approaches rely on calculating a response function (Kramers-Heisenberg formula) that is obtained from a perturbative analysis of scattering processes in the frequency domain. This requires a knowledge of a full set of eigenstates in order to account for all intermediate processes away from equilibrium, limiting the applicability to small tractable systems. In this work, we present an alternative paradigm, recasting the problem in the time domain and explicitly solving the time-dependent Schrödinger equation without the limitations of perturbation theory: a faithful simulation of the scattering processes taking place in actual experiments, including photons and core electrons. We show how this approach can yield the full time and momentum resolved Resonant Inelastic X-Ray Scattering (RIXS) spectrum of strongly interacting many-body systems. We demonstrate the formalism with an application to Mott insulating Hubbard chains using the time-dependent density matrix renormalization group method, which does not require a priory knowledge of the eigenstates and can solve very large systems with dozens of orbitals. This approach can readily be applied to systems out of equilibrium without modification and generalized to other spectroscopies.

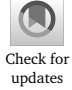

# 1  Introduction

Recent advances in experiments with light have paved the way to a new age in the study of elementary excitations of correlated matter [1–5]. High intensity X-ray sources, ultrafast pulses, and detectors with enhanced resolution for photon scattering measurements [1, 4, 6] have driven a continuous improvement of techniques such as X-ray absorption (XAS) and emission (XES) spectroscopies [7–9], resonant inelastic X-ray scattering (RIXS) [1, 4, 6, 10] as well as their corresponding dynamical versions (e.g. non-equilibrium or NE-XAS and time-resolved tr-RIXS). In particular, the possibility to probe with energy and momentum resolution excitations arising from charge, spin and orbital degrees of freedom has made RIXS the favorite tool to study the spectrum of solids and complex materials, including transition-metal compounds [11–15], Mott and anti-ferromagnetic insulators and unconventional high $T_c$ superconductors [16–21].

This fruitful period has also been marked by theoretical efforts to understand more in depth the scattering processes and the nature of the dynamical correlation functions probed by these experiments [1]. In this respect, uncovering various aspects underlying the excitation spectrum of a system is associated to the calculation of dynamical correlation functions, a task that, to date, remains challenging. The limitations of available techniques to compute spectral properties in strongly correlated systems away from equilibrium has curbed further theoretical progress [22]. For instance, the Bethe Ansatz [23] and Dynamical Mean-Field Theory (DMFT) [24] are restricted to relatively simple model Hamiltonians, whereas time-dependent Density Functional Theory (TD-DFT) [8] covers weakly coupled regimes. Exact digonalization (ED), which has been the most employed numerical tool to calculate of the spectrum of solids and complex materials [10–14, 25–39], provides access to small clusters and limits the momentum resolution.

The main limiting factor in these calculations is that core hole spectroscopies such as RIXS involve intermediate processes that can only be accounted for with an explicit knowledge of all the eigenstates of the system, requiring a full diagonalization of the Hamiltonian. Recently, Nocera *et al.* introduced a novel framework [40] based on the dynamical density matrix renormalization group (dDMRG) [41–43] aiming at extending the range of RIXS (and XAS) computations to systems beyond the reach of exact diagonalization. Even though cluster sizes much bigger than ED were reached, the algorithm proposed in Ref. [40] requires a number of DMRG simulations scaling linearly in the size of the system, making the computation of the entire RIXS spectrum a difficult task for challenging Hamiltonians.

In this work, we propose an alternative approach in which the calculation of spectrum is recast as a scattering problem that can be readily solved by means of the time-dependent DMRG method in a framework that does not require a full set of eigenstates of the Hamiltonian

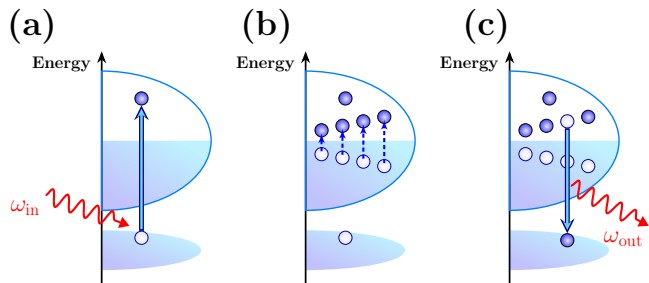

Figure 1: In a RIXS experiment, a core electron is excited into the valence band (a). After some time, an electron decays back into the core orbital, emitting a photon and leaving the system in an excited state, (b) and (c).

of the system. As we describe below, we explicitly introduce incident photons and core orbitals in the problem and numerically solve for the time evolution of the system accounting for the photon absorption and spontaneous emission in real time. This powerful formulation overcomes all the hurdles imposed by previous methods that work in the frequency domain and rely on explicitly obtaining dynamical spectral functions by means of generalized Fermi golden rules (Kramers-Heisenberg formula) in the frequency domain. The idea of simulating the problem in real time instead of carrying out notoriously difficult calculations in the frequency domain have been tested in the simpler context of photoemission [44] and neutron scattering [45]. We hereby generalize them to the much more complex case of X-ray spectroscopies.

The paper is organized as follows: in Sec. 2 we review the principles of light-matter interactions taking place in X-ray experiments. With this foundation, we then introduce our approach and show how to recast the calculation of the spectrum as a time-dependent scattering problem. We describe a practical implementation for RIXS in Sec. 3. We present results using the time-dependent density matrix renormalization group method (tDMRG) [46–49] in Sec. 4 for a one-dimensional Mott insulator described by a Hubbard chain, a model which has been widely used to simulate RIXS in cuprates. Finally, we discuss our findings and implications in Sec. 5.

## 2 Light-matter interactions

We hereby briefly review the basic ideas describing an X-ray scattering experiment and provide a theoretical background to put the problem in context. Since X-rays are a highly energetic beam of photons, we represent them through a vector potential:

$$\mathbf{A}(\mathbf{r}) = \sum_{\mathbf{k},\lambda} A_{\mathbf{k}} \mathbf{e}_{\mathbf{k}\lambda} (b_{\mathbf{k}\lambda} e^{i\mathbf{k}\mathbf{r}} + b_{\mathbf{k}\lambda}^{\dagger} e^{-i\mathbf{k}\mathbf{r}}), \tag{1}$$

where $A_{\mathbf{k}} = \sqrt{2\pi c^2 / V_s \omega_{\mathbf{k}}}$ is the normalized amplitude in volume $V_s$, with $\omega_{\mathbf{k}} = c|\mathbf{k}|$. The polarization unit vectors $\mathbf{e}_{\mathbf{k}\lambda}$ ($\lambda$=1,2) point in directions perpendicular to the propagation of the photons with momentum $\mathbf{k}$, represented by the conventional bosonic creation and annihilation operators $b^{\dagger}$, $b$. The full Hamiltonian including the solid and the radiation field is written as

$$H = H_0 + H_{ph} + V, \tag{2}$$

where $H_0$ describes the electrons and nuclei in the solid, and $H_{ph} = \sum_{\mathbf{k},\lambda} \omega_{\mathbf{k}} (b_{\mathbf{k},\lambda}^{\dagger} b_{\mathbf{k},\lambda} + 1/2)$. The light-matter interaction is given by

$$V = \frac{e}{mc} \sum_i \mathbf{p}_i \cdot \mathbf{A}(\mathbf{r}_i) + \frac{e}{2mc} \sum_i \sigma_i \cdot \nabla \times \mathbf{A}(\mathbf{r}_i), \tag{3}$$

where the first term accounts for the interaction of the electric field with the momentum $\mathbf{p}$ of the electrons and the second term describes the magnetic field acting on the electron spin $\sigma$. In the following, we will ignore the magnetic interaction as well as higher order terms that are not included in this expression. Therefore, replacing the quantized vector operator (1) leads to:

$$V = \frac{e}{mc} \sum_i \sum_{\mathbf{k},\lambda} A_{\mathbf{k}} (b_{\mathbf{k},\lambda}^\dagger \mathbf{e}_{\mathbf{k},\lambda} \cdot \mathbf{p}_i e^{i\mathbf{k}R_i} + \text{h.c.}) \tag{4}$$

$$= \sum_{\mathbf{k},\lambda} (b_{\mathbf{k},\lambda}^\dagger D_{\mathbf{k},\lambda} + \text{h.c.}), \tag{5}$$

where we have assumed the dipole limit in which $e^{i\mathbf{k}\cdot\mathbf{r}} \simeq e^{i\mathbf{k}\cdot\mathbf{R}_i}$ where $\mathbf{R}_i$ is the position of the ion to which electron $i$ is bound, and we have introduced the dipole operator

$$D_{\mathbf{k},\lambda} = \sum_i \frac{e}{mc} A_{\mathbf{k}} \mathbf{e}_{\mathbf{k},\lambda} \cdot \mathbf{p}_i e^{i\mathbf{k}\cdot\mathbf{R}_i} . \tag{6}$$

We start our discussion of the scattering processes by first considering a case in which a photon with momentum $\mathbf{k}$, energy $\omega_{\mathbf{k}}$ and polarization $\mathbf{e}_{\mathbf{k},\lambda}$ is absorbed, leaving the system energetically excited. The possible final states will be determined by the allowed dipolar transitions. In particular, one finds:

$$\langle n'l'm'|\mathbf{p}_i|nlm\rangle \neq 0 \iff \Delta l = \pm 1, \quad \text{and} \quad \Delta m = 0, \pm 1,$$

where $\Delta l = l' - l$ and $\Delta m = m' - m$, and $l$ represents the orbital angular momentum with projection $m$. In the process we are interested in, this operator will create a core-hole excitation. In particular, we focus on the Cu $L$-edge ($2p \to 3d$) transition in a typical X-ray scattering experiment on a transition metal oxide cuprate material [50]. In this case,

$$D_{\mathbf{k},\lambda} = \sum_{i,\sigma,\alpha} (e^{i\mathbf{k}\cdot\mathbf{R}_i} \Gamma_\alpha^\lambda d_{i,\sigma}^\dagger p_{i,\alpha,\sigma} + \text{h.c.}), \tag{7}$$

where $d^\dagger$ adds an electron to the valence band ($3d_{x^2-y^2}$) and $p_\alpha$ creates a hole in a $2p_\alpha$ orbital. The coefficients $\Gamma_\alpha^\lambda$ are determined by the matrix elements of the dipole operator, $\Gamma_\alpha^\lambda \propto \langle 2p_\alpha|\mathbf{e}_{\mathbf{k},\lambda}\cdot\mathbf{r}|3d_{x^2-y^2}\rangle = 1$, where we have expressed the dipole operator in terms of the position operator $\mathbf{r}$ [1]. It is typically assumed that the core hole is strongly localized and only one Cu $2p_\alpha$ orbital is involved in the process.

Due to the large local spin-orbit coupling in the core $2p$ orbital (of the order of 20eV at the Cu L-edge) the $2p$ orbitals split by their total angular momentum $\tilde{j}$, corresponding to the $L_2$ ($\tilde{j} = 1/2$) and $L_3$ ($\tilde{j} = 3/2$) transition edges. Because the energy separation between the two resonances is much larger than the core-hole lifetime broadening, we neglect the possibility of interference between the two edges, such that we have either $D_{\mathbf{k}} \simeq D_{\mathbf{k},\tilde{j}=1/2}$ or $D_{\mathbf{k}} \simeq D_{\mathbf{k},\tilde{j}=3/2}$. As a consequence, neither the spin nor the orbital angular momentum of the $2p$ band are good quantum numbers in the scattering process, but only the total angular momentum is conserved, allowing for orbital and spin "flip" processes at the Cu-L edge RIXS [10,50]. For this reason, at the $L_3$ edge, in the following we approximate the dipole operator $D_{\mathbf{k},\lambda} = \sum_{i\alpha,\sigma}(e^{i\mathbf{k}\cdot\mathbf{R}_i} \Gamma_\alpha^\lambda d_{i,\sigma}^\dagger p_{i\alpha,\sigma} + \text{h.c.})$ as $D_{\mathbf{k}} = \sum_{i,\sigma,\sigma'}(e^{i\mathbf{k}\cdot\mathbf{R}_i} \Gamma^{\sigma,\sigma'} d_{i,\sigma}^\dagger p_{i,\sigma'} + \text{h.c.})$, where $\Gamma^{\sigma,\sigma'}$ contains Clebsch-Gordan coefficients for the $\tilde{j} = 3/2$ state and we also ignore polarization effects.

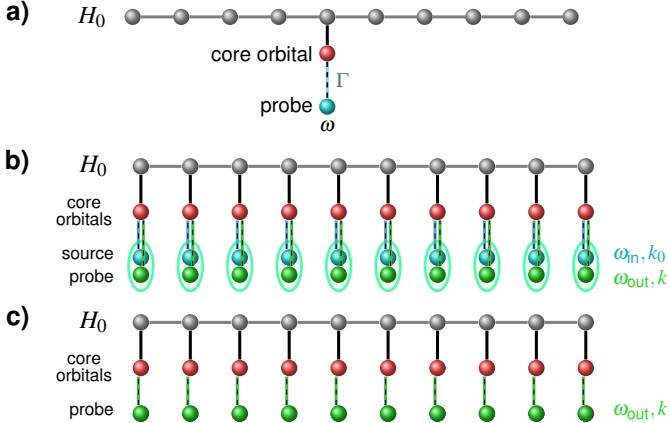

Figure 2: Schematic geometries used in the numerical calculations: (a) The local spectrum is measured adding one core orbital and emitted photon; (b) The momentum resolved RIXS calculation requires translational invariance, and an extended "detector" with one source and probe photon orbital per site; (c) in the semiclassical approach, the source photon is replaced by a classical field.

After the excitation is created, the conduction electrons will experience a local Coulomb potential $-U_c$ in the presence of the core hole and Hamiltonian (2) is modified accordingly:

$$H = H_0 + H_c + H_{ph} + V, \qquad H_c = -U_c \sum_i n_{di}(1 - n_{pi}),$$

$$V = V_{in} + V_{out}, \qquad V_{out} = V_{in}^\dagger,$$

$$V_{in} = \sum_k b_\mathbf{k} D_\mathbf{k} = \sum_\mathbf{k} b_\mathbf{k} \sum_i e^{i\mathbf{k}\cdot\mathbf{R}_i} D_i,$$

(8)

with $D_i = \sum_{\sigma,\sigma'} \Gamma^{\sigma,\sigma'} d_{i,\sigma}^\dagger p_{i,\sigma'}$ and $n_p = \sum_\sigma p_\sigma^\dagger p_\sigma$, and it is important to notice that the $p$ orbital can only be double or single occupied, but never empty since that would imply a two photon process.

## 3 Time-resolved RIXS

Resonant inelastic X-ray scattering (RIXS) is a high order process that can be described as a combination of X-ray absorption (XAS) and X-ray emission spectroscopy (XES), in which the system absorbs a photon with energy $\omega_{in}$ and momentumm $\mathbf{k}_{in}$ and emits another one with energy $\omega_{out}$, momentum $\mathbf{k}_{out}$. We hereby focus on the so-called "direct RIXS" processes, see Fig.1 and Fig. 1 in Ref. [12]). As a consequence, the photon loses energy $\Delta\omega = |\omega_{out} - \omega_{in}| = \omega_{in} - \omega_{out}$ (from now on referred-to as simply $\omega$) and the electrons in the solid end up in an excited state with momentum $\mathbf{k}_{out} - \mathbf{k}_{in}$. In the following, we consider $\mathbf{k}_{in} = 0$ and refer to the momentum transferred simply as $\mathbf{k}$. While in principle the resulting spectrum is a function of two frequencies, the incident photon $\omega_{in}$ is tuned to match one of the absorption edges, hence the resonant nature of the process. The final response is then determined by measuring the final occupation of the $\omega_{out}$ mode.

RIXS can be formulated in terms of a single photon being absorbed or emitted by the system. We consider the system locally connected to two photon orbitals, one with energy $\omega_{in}$ that will serve as the "source" for absorption, and a second one with energy $\omega_{out}$ will be the "detector" for emission and is initially empty. Since we are interested in obtaining

the momentum resolved spectrum we need extended probe and sources [44]. This can be implemented in two different but equivalent setups, that we proceed to describe below.

## 3.1 Photon-in, photon-out description

In this setup, illustrated in Fig.2(b), each site will have a core-orbital, a source orbital, and a probe or detector orbital. Since the system is translational invariant, total momentum will be conserved. The problem is described by the Hamiltonian:

$$
\begin{aligned}
H &= H_0 + H_c + \omega_{in} \sum_i n_{b,s,i} + \omega_{out} \sum_i n_{b,d,i} + V \,, \\
V &= V_{in} + V_{out} \,, \quad V_{out} = V_{in}^\dagger \,, \\
V_{in} &= \sum_{\sigma,\sigma'=\uparrow,\downarrow} V_{in}^{\sigma\sigma'} \,, \\
V_{in}^{\sigma\sigma'} &= \sum_i (\Gamma_s^{\sigma\sigma'} b_{s,i} + \Gamma_d^{\sigma\sigma'} b_{d,i}) d_{i\sigma'}^\dagger p_{i\sigma} \,, \\
H_c &= -U_c \sum_{i\sigma} (1 - n_{pi\sigma}) n_{di} \,,
\end{aligned}
\tag{9}
$$

where we have introduced couplings $\Gamma_{s/d}^{\sigma\sigma'}$ that can be turned on and off selectively depending of the case of interest, as we describe below. For instance, in the absence of spin-orbit interaction, only $\Gamma_{s/d}^{\sigma\sigma}$ will be non-zero. Otherwise, the spin projection is no longer a good quantum number and the core-electron is allowed to flip spin when it is excited.

At time $t = 0$, the problem is initialized with the core-orbitals double-occupied, the probe orbitals empty, and a single source photon with momentum $\mathbf{k}_{in}$. This can be done by means of a projector:

$$
H_{source} = -|\mathbf{k}_{in}\rangle\langle\mathbf{k}_{in}| + \lambda \sum_{ij} n_{b,s,i} n_{b,s,j} \,,
\tag{10}
$$

where $|\mathbf{k}_{in}\rangle\langle\mathbf{k}_{in}| = n_{b,s}(\mathbf{k}) = \frac{1}{L} \sum_{mn} e^{i\mathbf{k}_{in}(\mathbf{R}_m - \mathbf{R}_n)} b_{s,m}^\dagger b_{s,n}$ and the second term multiplied by a large positive constant $\lambda > 0$ represents a boson-boson repulsion that ensures that there is only one photon overall. This is necessary because the total Hamiltonian does not conserve photon number. We observe that since there is only one photon at play, only one core-electron will be excited at most at any given time. The full calculation proceeds as follows: The system is first initialized in the ground state of $H_0 + H_{source}$. The energy $\omega_{in}$ is set to the transition edge and tDMRG simulations are carried out in parallel for each value of $\omega_{out}$. The full spectrum is obtained by measuring the momentum distribution function of the detector at time $t_{probe}$, $n_{b,d}(\mathbf{k}) = 1/N^2 \sum_{mn} e^{i\mathbf{k}(\mathbf{R}_m - \mathbf{R}_n)} b_{d,m}^\dagger b_{d,n}$.

## 3.2 Semi-classical description

In order to reduce the computational complexity of the problem, we hereby introduce a semi-classical approach to completely eliminate the source degrees of freedom. This alternative approach prescinds from the incoming photon and source orbitals and reduces the cost of the simulation exponentially.

The premise relies on the fact that –unlike spontaneous emission– the absorption process can be described semi-classically without using quantum photons by means of a classical field

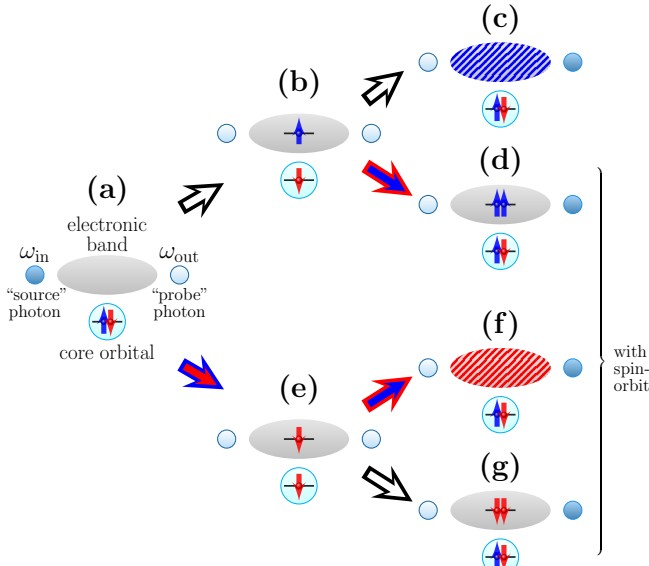

Figure 3: Depiction of the possible processes accounted for by our formulation of the direct RIXS problem: The incident photon has energy $\omega_{in}$, and the "detector" is tuned to a target energy $\omega_{out}$. The color arrows indicate processes in which a spin flip is involved, due to the spin orbit term. The (b)→(c) path represents a process without spin orbit in which the final state of the band has the same quantum numbers as the original one. The (e) path corresponds to a spin-flip after absorption, while (d) and (f) undergo a spin-flip after emission. The final state can reverse the spin flip (d),(f) or leave the band with a different spin (c),(g). There is a similar cascade of processes related to these by time-reversal.

coupled to the dipole operator or, equivalently, in the form of a "gate potential":

$$H = H_0 + H_c + \omega_{out}\sum_i n_{b,d,i} + \omega_{in}\sum_i n_{p,i} + V\,,$$

$$V = V_{in} + V_{out}\,, \quad V_{out} = V_{in}^\dagger\,,$$

$$V_{in} = \sum_{\sigma,\sigma'=\uparrow,\downarrow} V_{in}^{\sigma\sigma'}\,,$$

$$V_{in}^{\sigma\sigma'} = \Gamma_d^{\sigma\sigma'}\sum_l b_{d,l}d_{l\sigma'}^\dagger p_{l\sigma} + \Gamma_s^{\sigma\sigma'}\sum_l d_{l\sigma'}^\dagger p_{l\sigma}\,,$$

$$H_c = -U_c\sum_{i\sigma}(1-n_{pi\sigma})n_{di}\,, \tag{11}$$

where the "gate voltage" $\omega_{in}$ acts on all the core electronic states and we include a "hopping" term between the core states $p$ and the conduction states $d$. The corresponding setup is illustrated in Fig.2(c). Notice that this is a modification of the chain geometry, where the incoming photons are replaced by the voltage term. In our calculations we take $\omega_{in}$ corresponding to the XAS transition edge. As time evolves, due to energy and momentum conservation, core electrons with energy $\omega_{in}$ will be able to "tunnel" to the conduction/valence band, same as before. It can be easily shown that both formulations are mathematically equivalent, something that we have also corroborated numerically (not shown): results from the quantum and semi-classical approaches are indistinguishable.

An important consideration to take into account is that, since we are interested in a single photon process, only one core electron can be excited to the conduction band at a time. This

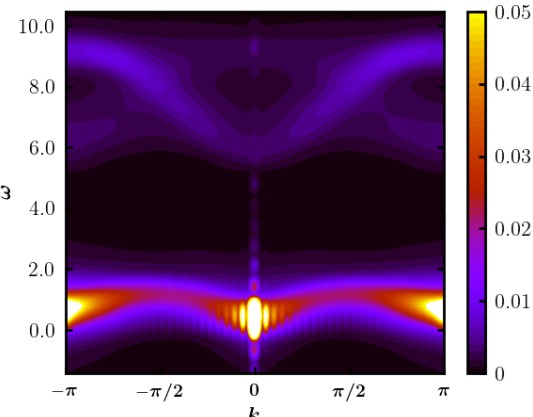

Figure 4: Momentum resolved RIXS spectrumn for a Hubbard chain with $U = 8$, $U_c = 1.5$, $\Gamma = 0.2$ and $L = 64$ sites, showing only the direct spin conserving channel using a Peierls phase replacing the source photon for the absorption process.

is naturally accounted for by the quantum mechanical formulation in Sec.3.1 with the source photons explicitly included. Therefore, the core orbitals can never be found empty, and only single and double occupied states are allowed. In practice, this can be done by either not including these states in the basis, or by adding a very large energy cost to states with empty core orbitals.

### 3.3 Measurement protocol

The goal is to measure the momentum distribution function of the detector $\langle \psi(t) | n_{b,d}(\mathbf{k}) | \psi(t) \rangle$ after the scattering term is turned on (see Fig.2). We hereby focus on the implementation using the semi-classical description. Explicitly, one proceeds by carrying out a time-dependent simulation with the probe orbitals empty, and the core orbitals double occupied. The initial state of the electronic system may or may not be an eigenstate and could be very far from equilibrium, since this approach applies regardless of the case. By an appropriate choice of $\Gamma_s^{\sigma\sigma'} = \Gamma_d^{\tau\tau'} = \Gamma$, and all others set to zero, one evolves the system in time to obtain a wavefunction $|\psi_{\sigma\sigma',\tau\tau'}(t)\rangle$. This allows us to resolve the different contributions to the spectrum that split into spin conserving and non-conserving ones:

$$I_{RIXS}^{\Delta S^z = 0} = \langle \psi_{\uparrow\uparrow,\uparrow\uparrow} | n_{b,d} | \psi_{\uparrow\uparrow,\uparrow\uparrow} \rangle, \tag{12}$$

$$I_{RIXS}^{interference} = \langle \psi_{\uparrow\uparrow,\uparrow\uparrow} | n_{b,d} | \psi_{\downarrow\downarrow,\downarrow\downarrow} \rangle, \tag{13}$$

$$I_{RIXS}^{\Delta S = 1} = \langle \psi_{\uparrow\uparrow,\downarrow\uparrow} | n_{b,d} | \psi_{\uparrow\uparrow,\downarrow\uparrow} \rangle. \tag{14}$$

The first expression, Eq.(12) involves the path (a)-(b)-(c) in Fig.3, while Eq.(13) the "interference" between this process and the one related by time-reversal symmetry with the "down" electron. The term Eq.(14) originates from the (a)-(b)-(d) channel, that results in a net spin flip in the conduction band in the presence of spin-orbit coupling. It is easy to see that these are the only distinct contributions, since all others are related by time-reversal symmetry.

By turning the couplings $\Gamma$ on and off, this protocol allows us to measure each of the terms individually, including the interference Eq.(13) and the spin-flip contributions in Eq.(14). To account for all the terms, a total of four independent calculations would be required (notice that the interference term (13) involves two different wave-functions). However, setting all the $\Gamma_s^{\sigma\sigma'} = \Gamma_d^{\sigma\sigma'} = \Gamma$ will yield the total RIXS spectrum automatically from a single time-dependent simulation.

# 4 Results

We hereby demonstrate how to implement our protocol using the tDMRG method [46–49]. We describe a $d$ band by means of the Hubbard model in one-dimension with open boundary conditions:

$$H = -J \sum_{i=1,\sigma}^{L-1} \left( d_{i\sigma}^\dagger d_{i+1\sigma} + \text{h.c.} \right) + U \sum_{i=1}^{L} n_{di\uparrow} n_{di\downarrow}. \tag{15}$$

Here, $d_{i\sigma}^\dagger$ creates an electron of spin $\sigma$ on the $i^{\text{th}}$ site along a chain of length $L$. The on-site Coulomb repulsion is parametrized by $U$, while we express all energies in units of the hopping parameter $J$ (the symbol "$t$" will be reserved to represent time, which will be expressed in units of $1/J$). The geometry of the problem is illustrated in Fig.2(a). We consider a half-filled band describing a one-dimensional Mott insulator.

We simulated a chain of length $L = 64$ at half-filling ($N = L$ electrons), $U = 8$, $U_c = 1.5$ and evolved in time up to $t = 15$ and time steps $\delta t = 0.1$, which yields a truncation error smaller than $10^{-7}$. Since the terms in the Hamiltonian are local, we can use a Suzuki-Trotter decomposition of the time-evolution operator.

## 4.1 Momentum resolved RIXS

With the probe extended to the entire volume of the system, we resolve the momentum dependence of the spectrum, as seen in Fig.4 (We show only results for the "direct" term, $I_{RIXS}^{\Delta S^z=0}$, Eq.(12)). We use the semi-classical approach (without source photons) and take $U = 8$, $U_0 = 1.5$. We tune $\omega_{in} = 4.47$ to the transition edge for this value of $U$ and use 200 states. After obtaining the ground state we connect the extended probe at time $t = 0$ and measure the momentum distribution function $n_{b,d}(k,t)$ at the detector as a function of time. Since in our tDMRG simulations we use open boundary condition, the proper definition of momenta corresponds to particle in a box states $\sin(k_j x)$ with momenta $k_j = j\pi/(L+1)$ with ($j = 1, \cdots, L$). However, as customary in DMRG calculations, we vary $k$ continuously.

The dispersive features clearly allow us to identify different contributions. First, we note a bright low energy band that resembles the standard two-spinon continuum characterizing the low energy magnetic excitations spectrum of antiferromagnetic 1D spin-chain cuprates, such as $Sr_2CuO_3$ and $SrCuO_2$. These have been experimentally investigated both using RIXS at the Cu L-edge [50] and neutron scattering spectroscopy [51, 52]. Particularly evident is the sharp, large intensity peak at $k = 0$, $\omega = 0$ which represents the elastic peak, typical of RIXS experiments. We note that while this peak can easily be subtracted off in RIXS dDMRG calculations [53], this cannot be done in our tunneling approach, except in an ad-hoc way.

Finally, we highlight the upper excitation band at $\omega \in [6t, 10t]$, centered around $\omega \simeq 8t$, which describes particle hole (holon-doublon) excitations to the upper Hubbard band. Our results show a much better quality compared to dDMRG calculations at high energies, as in this case dDMRG precision deteriorates because it needs to converge over a big Krylov subspace [41, 54–56]. In particular, even though much less intense than the low energy spin excitation band, our approach can clearly resolve interesting features of the holon-doublon band, which can be fit by a cosine-like dispersion $E \simeq 8t - 2t\cos(k)$, plus an incoherent background at large momentum transfers. Our approach can therefore be of great help in the interpretation of RIXS experiments in low-dimensional cuprates where, besides magnetic excitations, also charge excitations play an important role, such as in correlated charge-density wave states [57, 58].

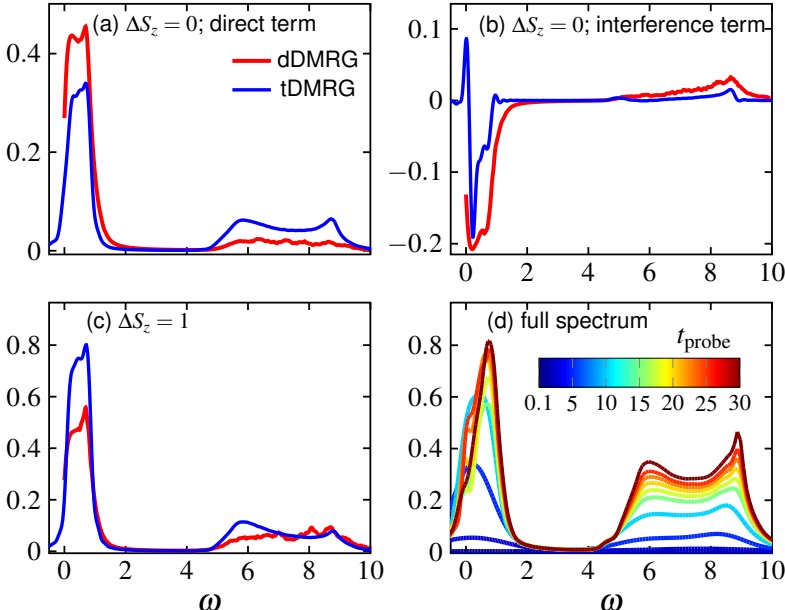

Figure 5: RIXS results using a time-dependent scattering approach for a Mott insulating Hubbard chain of length $L = 64$ with $U = 8$ and $U_c = 1.5$ and $t_{probe} = 30$. Panels show (a) the direct contribution, (b) interference term, (c) spin-flip term, and (d) full spectrum, also following its evolution as a function of time. dDMRG results are also shown for comparison.

## 4.2 Single site spectrum

In order to gain more insight on the features and contributions at different energy scales, we now focus on a single site setup to calculate the local RIXS spectrum for the Mott insulating Hubbard chain. In this case, we use the geometry depicted in Fig.2(a), with a single core orbital and detector. Results obtained using the time-dependent scattering approach with $m = 300$ DMRG states are shown in Fig.5, compared to data obtained with dDMRG for the same system size and $m = 800$ states. We show results for the different contributions to the spectrum, Eq. 12 and Eq. 14, and also the full spectrum with all couplings $\Gamma^{\sigma\sigma'} = \Gamma$. We notice in Fig.5(d) that the lineshape "evolves" as a function of probing time, with the different features becoming better resolved at longer time. This is a manifestation of the time-energy uncertainty principle. In fact, experiments are usually poorly resolved due to the ultrafast nature of them and the short core-hole lifetime. We also point out that the elastic contribution at $\omega = 0$ has been subtracted from the dDMRG data, but not from the tDMRG results. The overall agreement is qualitatively very good, but the dDMRG results clearly suffer from poor precision particularly at high energies, as pointed out above. The discrepancies can be attributed to the fact that the time-evolved wave-function contains higher order contributions.

In agreement with the momentum resolved features described earlier, the resulting spectrum contains signatures of a broad high-energy band, and a narrower low energy band with larger concentrated weight. Since the elastic contribution has been removed from the dDMRG data, this low energy band indeed corresponds to spectral weight *in the gap*. While this can be confusing, it is readily explained in terms of multi spinon excitations with $\Delta S_z = 0$ produced by even number of spin flips and not changing the number of electrons [23]. While a Mott insulator has a charge gap, the spin excitations are gapless, a manifestation of spin-charge separation in one spatial dimension. Clearly, there are no available states within the Mott gap. The high energy features correspond to holon-doublon excitations that transfer spectral

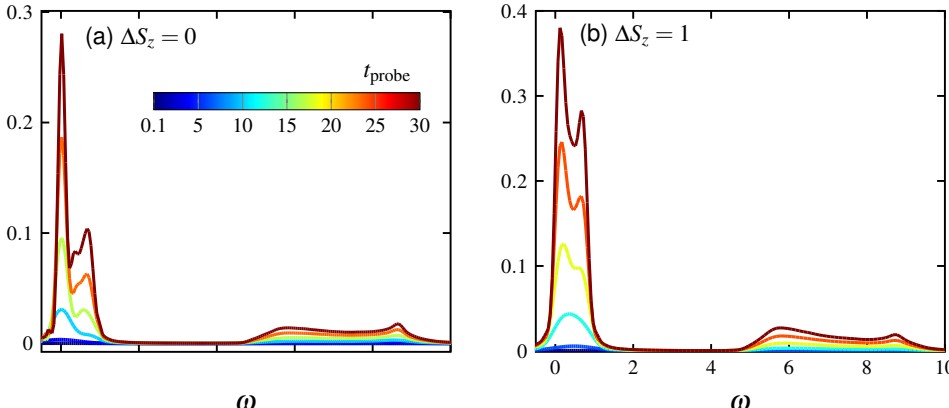

Figure 6: (a) Spin conserving and (b) non-conserving RIXS channels for a Mott insulating Hubbard chain with $U = 8$, $U_c = 1.5$, $\Gamma = 0.1$, $L = 64$ sites, for different probing times $t_{probe}$. The inelastic features at low energies emerge after a characteristic time for the electrons to break and decay into spinons and doublons.

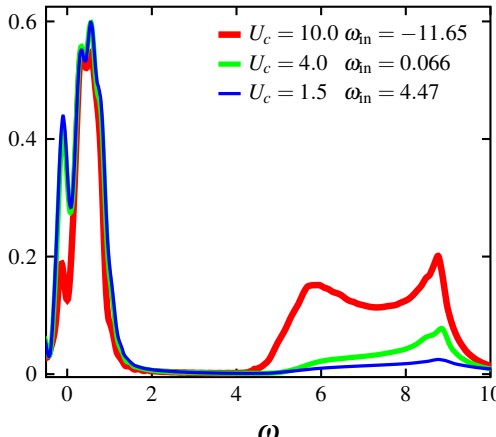

Figure 7: RIXS results (without spin-orbit interaction) for a Mott insulating Hubbard chain with $U = 8$, $L = 64$ sites, and different values of the core-hole potential $U_c$ measured at time $t_{probe} = 30$.

weight from the lower into the upper Hubbard band. In a Mott insulator away from equilibrium one can imagine high-order processes in which an electron in the band decays into the core orbital, simultaneously creating a particle hole excitation with an additional doublon in the upper Hubbard band. The decay of the doublon into spinons is unlikely, due to the weak spinon dispersion and the vanishing coupling between charge and spin [59–61](long doublon lifetime has also been observed in higher dimensions [62–65]).

## 4.3 Core-hole lifetime and dependence of the lineshape

The core-hole lifetime is in general controlled by non-radiative decay mechanisms due to electronic correlations, such as Auger decay. Even though it is believed that the core-hole lifetime is an intrinsic property of atom, it was recently demonstrated that it can be tuned in certain cases by employing thin films x-ray planar cavities [66].

Unlike dDMRG, where the core-hole lifetime is introduced as an artificial damping in the Kramers-Heisenberg formula, in our approach the core-hole lifetime is an intrinsic property of the problem and is determined by the magnitude of the light-matter interac-

tion $\Gamma$. In particular, the inverse core-hole *bare* linewidth for radiative decay is given by $\hbar\tau_{\text{core-hole}} = \pi\sum_{\mathbf{k}}|\Gamma_{\mathbf{k}}|^2\delta(\epsilon_{\mathbf{k}} - \epsilon_c - \omega_{in})$, where $\epsilon_{\mathbf{k}}$ is the energy of the valence band electrons while $\epsilon_c$ is the core-level energy. [2]. In fact, the lifetime of the core-hole, while short, is actually momentum and energy dependent, and is intrinsically accounted for by our formulation. In our calculations, the lineshape for a fixed $\Gamma$ will be determined by the time lapsing between the moment the photon source is turned on, and the emitted photon is measured by the detector, our $t_{probe}$ for a smaller value of $\Gamma$. In Fig.6 we show RIXS results as a function of the probing time $t_{probe}$. As seen in Fig.6, different features evolve with different characteristic times. At short times we observe the development of an elastic peak with little spectrum in the gap. This corresponds to an excitation being created at the transition edge and immediately recombining by emitting a photon, without energy loss and time to break into spinons, yielding only an elastic signal at $\omega = 0$. On the other hand, if we allow the system to evolve under the action of the core-hole potential between absorption and emission, the resulting attractive force will bind the doublon to the core orbital and create spin domain walls (spinons), that will propagate throughout the system. This is observed in the same figure, where the higher energy features in the spectrum evolve and become better resolved as we increase $t_{probe}$.

We point out that other factors that are not accounted by our formulation can play a role in the core-hole lifetime of actual materials, that can be very short due to non-radiative recombination mechanisms such as Auger decay [67, 68]. In our formulation, this effects could be included through a "core bath", as suggested in Ref. [69].

## 4.4 Dependence on the core-hole potential

Finally, we observe that the magnitude of $U_c$ affects the relative spectral weight between the high and low energy bands. This is demonstrated in Fig.7, where we show the RIXS spectra obtained by varying $U_c$ from 1.5 to 10. For large $U_c$, the core hole and a single doublon form a tightly bound state localized at the position of the core-orbital, effectively cutting the system in two. In the limit of $U_c \to \infty$, scattering with this potential induces holon-doublon excitations transferring weight into the upper Hubbard band, with a consequent increase in the spectral signal at high energies, while barely affecting the spectrum at low energies that originates from the spin degrees of freedom.

Our numerical results can be understood in direct analogy with those obtained in indirect RIXS [70]: given the very small recombination rate of the holon-doublon after the absorption process, our data strongly suggest that the core-hole-holon-doublon bound state has a lifetime larger than the *bare* core-hole one. In fact, the doublon state is playing the role of the *spectator* (unoccupied) conduction band level in standard indirect RIXS. Indeed, in indirect RIXS, by increasing the core-level interaction $U_c$, and therefore increasing the incident energy at resonance, the spectral weight of charge (holon-doublon) excitations increases relative to that one of spin excitations.

# 5 Conclusions

We have introduced a time-dependent scattering approach to core-hole spectroscopies that allows one to carry out numerical calculations without the full knowledge of the excitation spectrum of the system. The applicability of the method is demonstrated by means of time-dependent DMRG calculations. Results for the Hubbard model are achieved with minimal effort on large systems using a fraction of the states –and simulation time– required by the dDMRG formulation. Unlike dDMRG and other approaches such as ED and DMFT that rely on explicitly calculating dynamical response functions based on the Kramers-Heisenberg formula, our time-dependent calculations are not limited by perturbation theory and contain contribu-

tions from higher order processes. Momentum resolution is obtained by means of extended source and probe, and an exponential speed up can be obtained by modeling the absorption process semi-classically by means of an oscillating field or, equivalently, of a "gate potential" term. In addition, our time-dependent approach can be readily applied without modification to non-equilibrium situations in which the electronic band is not in the ground state.

Our method differs from others in one notorious aspect: while the general approach essentially consists of first calculating a spectrum corresponding to an infinite core-hole lifetime, and then convoluting this spectrum with a Lorentzian lifetime broadening, in our time-dependent simulations there is no way to control the internal dynamics of the system and the core-hole lifetime is basically determined by the probing time $t_{probe}$. At long times we always observe the emergence of low energy states that can be associated to gapless multi-spinon excitations.

In our calculations accuracy is kept it under control by using a sufficiently large number of DMRG states (bond dimension). Notice that at time $t = 0$ the system in in the ground state of $H_0 + H_{source}$. The core orbitals are in a product state of double occupied states and do not contribute to the entanglement. As time evolves, one electron will be excited from the core-orbitals, and one photon will eventually be emitted when the core hole recombines. When this occurs, the core orbitals return to a product state. Moreover, there is no hopping for the core degrees of freedom. This means that any additional entanglement will stem from the perturbations left behind in the system (which is a gapped Mott insulator) and the single photon at the detector, which will contribute to the entanglement by a bounded amount $\mathcal{O}(1)$. As a consequence, the entanglement growth will be minimal and simulations can proceed to quite long times. While we have not done a detailed quantitative analysis, we believe that the entanglement growth will be comparable, if not lower, than typical tDMRG simulations of spectral functions, particularly in the case of single-site RIXS.

Our formulation opens the door to the numerical study of core-hole spectroscopies in strongly correlated systems away from equilibrium using exact many-body approaches such as as tDMRG. The results obtained by these means give one access to transient regimes and allow one to resolve different excitation and decay mechanisms in real time. These ideas can easily be implemented within other numerical frameworks, such as time-dependent DMFT.

## Acknowledgments

We thank Shaul Mukamel for valuable feedback and Robert Markiewicz for stimulating discussions. We acknowledge generous computational resources provided by Northeastern University's Discovery Cluster at the Massachusetts Green High Performance Computing Center (MGHPCC). AN acknowledges the support from Compute Canada and the Advanced Research Computing at the University of British Columbia, where part of simulations were performed.

**Funding information**   AN is supported by the Canada First Research Excellence Fund. KZ was supported by a Faculty of the Future fellowship of the Schlumberger Foundation. AEF acknowledges the U.S. Department of Energy, Office of Basic Energy Sciences for support under grant No. DE-SC0014407.

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
