# Peer review of "A time-dependent momentum-resolved scattering approach to core-level spectroscopies"

_SciPost Physics, doi:SciPost Phys. 15, 166 (2023)_

## Round 1 · Referee Report · Anonymous (Referee 1) · 2023-3-14

Report

This paper proposed a new time-dependent scattering approach for numerically calculating core-hole spectroscopies such as resonant inelastic x-ray scattering (RIXS). As a demonstration, RIXS for a one-dimensional Hubbard model at half filling was calculated by employing time-dependent density-matrix renormalization group (tDMRG) based on the time-dependent scattering approach. Momentum dependence, core-hole potential dependence, and observing-time dependence of RIXS were demonstrated. The results demonstrated seem to be reasonable and give a hope for new theoretical progress in this field, for example, in time-dependent RIXS for photoexcited strongly correlated electron systems. Therefore, I think that this paper is worth publishing somewhere. However, before recommending in SciPost Physics, I request the authors to reconsider details of formulation and clarify physical meaning of the calculated results.

  1. The matrix element of the dipole operator $\Gamma_\alpha^\lambda$ in Eq. (7) may depend not only on orbital part but also on spin part. Below Eq. (7), the orbital part of the matrix element is expressed to be unity. However, this means that information on polarization is lost. Is this unity correct?
  2. The core 2$p_\sigma$ operator has indexes position $i$, orbital $\alpha$, and spin $\sigma$ in Eq. (7). However, the core 2$p_\sigma$ operator used to define its number operator below Eq. (8) does not have indexes about $i$ and $\alpha$. This is an inconsistent expression. Because of this inconsistency, I doubt $1-n_{pi}$ in $H_c$ of Eq. (8). Is this $2-n_{pi}$ and missing $\alpha$?
  3. In Eq. (8), there is no energy level of 2$p_\sigma$. It should be justified.
  4. There is no definition of $\mathbf{k}'$ and $\mathbf{k}$ in the first paragraph of Sec. 3. According to the definition of $\Delta\omega$, $\Delta\omega$ is negative. Is this correct definition? In Figs. 4-7, the horizontal axis is denoted by $\omega$. Is this equivalent to $\Delta\omega$? Also, what is the definition of $k$ in the horizontal axis in Fig. 4?
  5. What is the relation between $\sigma$ in the operator $p_{i\sigma}$ in Eq. (9) and the $z$ component of total angular momentum $\tilde{j}$?
  6. Is there information on polarization in Eq. (9)? Or does it already disappear from the model because of the item 1 above?
  7. According to the term of $\omega_{in}$ in Eq. (10), the energy of XAS becomes negative. Is this correct?
  8. There is no information on Fig. 2(c) in the main text.
  9. In the first line of page. 8, there is a statement that "all others set to zero". What are "all others"? They should be specified.
  10. Figure 4 contains both non-spin-flip and spin-flip contributions within the subspace of conserving the $z$ component of total spin. Therefore, $\Delta S=0$ should be $\Delta S_z =0$. Otherwise some confusions will appear among readers.
  11. There is no detailed explanation of each panel in Fig. 5 in the main text.
  12. Is the elastic contribution removed from tDMRG results in Fig. 5?

Requested changes

  1. Page 1: "(ED)"appears twice.
  2. Page 5, line 2: $j\rightarrow \tilde{j}$
  3. Page 7, Sec. 3.3, line 2: (see Fig. 2)$\rightarrow$ (see Fig. 3)
  4. Page 8, Sec. 4.1, line 3: $U_0=1.5\rightarrow U_c=1.5$

  • validity: -
  • significance: -
  • originality: -
  • clarity: -
  • formatting: -
  • grammar: -

Author:  Adrian Feiguin  on 2023-03-31  [id 3531]

(in reply to Report 2 on 2023-03-14)

We thank the Referee for a detailed report and a rigorous reading of the manuscript. In the following we proceed to addressing their comments, which have greatly helped us to make the manuscript more clear and readable.

1) . The matrix element of the dipole operator Γλα in Eq. (7) may depend not only on orbital part but also on spin part. Below Eq. (7), the orbital part of the matrix element is expressed to be unity. However, this means that information on polarization is lost. Is this unity correct?

Answer: We understand that the previous version of the manuscript wasn't clear on the notation and needed a revision. We have incorporated a number of changes that we believe have noticeably improved the rigor of the description and also made it more readable. Among them, the description of the dipole D operators and the couplings Gamma. Concerning specifically the Referee's question, we clarify that the dipole operators as defined in Eq.(7) do not flip the spin of the electrons. We point out that our description follows several of the works cited by us, such as Refs. [3-10]. The spin flip originates from the fact that the p orbitals have a strong spin-orbit coupling that splits the degenerate manifold by the total angular momentum J. That means that the p_{j=3/2} orbital/operators mix spin up and down, which in practice appears as a spin flip. We have clarified this and several other issues with the notation and also extended our discussion under Eq.(7) that will also address the additional questions by the Referee that follow. Concerning the polarization, the answer is yes, we ignore polarization effects, which we also clarify in the text.

2) The core 2pσ operator has indexes position i, orbital α, and spin σ in Eq. (7). However, the core 2pσ operator used to define its number operator below Eq. (8) does not have indexes about i and α . This is an inconsistent expression...

Answer: This follows our answer to 1). Indeed that notation was inconsistent, and we hope that the new version clarifies it and addresses this concern.

3) In Eq. (8), there is no energy level of 2pσ. It should be justified.

Answer: Follows 1). Indeed, the orbital index is spurious because of the spin-orbit coupling and we are interested in the L3 edge. The new version of the text, particularly following Eq.(7), addresses this and other issues with the notation.

4)There is no definition of k′ and k in the first paragraph of Sec. 3. According to the definition of Δω, Δω is negative. Is this correct definition? In Figs. 4-7, the horizontal axis is denoted by ω . Is this equivalent to Δω? Also, what is the definition of k in the horizontal axis in Fig. 4?

Answer: We thank the Referee for spotting these issues. We have clarified them in several parts of the text. In particular:

We have fixed the notation: "the system absorbs a photon with energy $\win$ and momentumm $\kin$ and emits another one with energy $\wout$, momentum $\kout$. We hereby focus on the so-called ``direct RIXS'' processes, see Fig.\ref{fig:fig2} and Fig.~1 in Ref.\cite{Kourtis2012-PhysRevB.85.064423}). As a consequence, the photon loses energy $\Delta \omega=|\wout-\win|=\win-\wout$ (from now one referred-to as simply $\omega$) and the electrons in the solid end up in an excited state with momentum $\kout-\kin$. In the following, we consider $\kin=0$ and refer to the momentum transferred simply as $\kk$."

5) What is the relation between σ in the operator piσ in Eq. (9) and the z component of total angular momentum j?

Answer: Follows 1). This comes from the dipole operator written using the Clebsch-Gordan coefficients for J=3/2, as we discuss after Eq.(7). This is also explained in the last sentence of Sec. 2: "As a consequence, neither the spin nor the orbital angular momentum of the $2p$ band are good quantum numbers in the scattering process, but only the total angular momentum is conserved, allowing for orbital and spin ``flip'' processes at the Cu-L edge RIXS\cite{schlappa2012spin,Jia2016-PhysRevX.6.021020}. "

6) Is there information on polarization in Eq. (9)? Or does it already disappear from the model because of the item 1 above?

Answer: Indeed, the polarization information is lost. We removed the corresponding label lambda that was spurious and just adding to the confussion.

7) According to the term of ωin in Eq. (10), the energy of XAS becomes negative. Is this correct?

Answer: The energy win is 4.47 for the value of U=8 and Uc=-1.5 used here, although we also considered other cases as shown in Fig. 7. Indeed, sometimes it can be negative. Notice that this energy is relative to the unperturbed ground state energy, and the fact that it can be negative correspond to the formation of a localized tightly bound state that shifts the energies effectively like a chemical potential.

8) There is no information on Fig. 2(c) in the main text.

Answer: We thank the Referee once again for this attention to detail. We have referred to it in the appropriate location in the text.

9) In the first line of page. 8, there is a statement that "all others set to zero". What are "all others"? They should be specified.

Answer: We are saying that by picking a particular choice of "Gammas" we can obtain different wave functions. By using these different wave functions, we can thus obtain all the independent contributiond to the spectrum, as describe din the text. We have rephrased this sentence, hoping to improve clarity:

"By an appropriate choice of $\Gamma_s^{\sigma\sigma'}=\Gamma_d^{\tau\tau'}=\Gamma$, and all others set to zero, one evolves the system in time to obtain a wave-function $|\psi_{\sigma\sigma',\tau\tau'}(t)\rangle$. This allows us to resolve the different contributions to the spectrum that split into spin conserving and non-conserving ones:..."

  1. Figure 4 contains both non-spin-flip and spin-flip contributions within the subspace of conserving the z component of total spin. Therefore, ΔS=0 should be ΔSz=0. Otherwise some confusions will appear among readers.

Answer: We appreciate the observation. We have fixed the notation both in the text and the figures.

11) There is no detailed explanation of each panel in Fig. 5 in the main text.

Answer: We have expanded our description of Fig.5 in the next. We point out that these curves are shown in order to compare to DDMRG data, and their physical interpretation is similar in all three cases (namely, multi-spinon processes at low energies, and doublon physics above the gap), and do not require separate case by case analysis.

12) Is the elastic contribution removed from tDMRG results in Fig. 5?

Answer: It is not. This is addressed in the sentece:

"Particularly evident is the sharp, large intensity peak at $k=0$, $\omega=0$ which represents the elastic peak, typical of RIXS experiments. We note that while this peak can easily be subtracted off in RIXS dDMRG calculations\cite{Nocera2018}, this cannot be done in our tunneling approach, except in an ad-hoc way."

We have also fixed the typos found by the Referee (We believe that the sentence concerning Fig. 2 actually points to the right figure, as it was intended by us)

---

## Round 1 · Referee Report · Anonymous (Referee 2) · 2023-3-17

Strengths

  • a new scalable MPS-based method for computing RIXS spectra in 1-D correlated materials
  • benchmarks on the 1-D Hubbard models, showing the performance of the method

Weaknesses

  • the explanations are not always clear; even a DMRG/MPS practicioner has to work hard to figure out the details of the computational scheme.

Report

In this paper, the authors introduce a new method for computing the RIXS spectrum for one-dimensional systems using time-dependent DMRG. RIXS is an experimental technique that is currently gaining more attention for probing the low-energy spectrum of correlated materials, so new computational methods are certainly wanted in this context. In contrast to spectral functions in ARPES or spin structure factors in inelastic neutron scattering, the RIXS spectrum is not readily available by "standard" MPS-based methods, and only recently an extension of dynamical DMRG has been proposed to capture this. In this respect, the current work is certainly interesting as it seems to yield access to RIXS spectra in a scalable way for 1-D systems.

In this sense, I believe that the paper deserves publication. However, I think that the paper would benefit from some clarifications before publication.

Requested changes

As I see it, the method originates from earlier works by some of the same authors, and much of the technical matters are not really discussed here. This makes the paper less readable, and I would like to get a few clarifications from the authors: 1) Is the method, in fact, a straightforward continuation of these earlier works? It would be good to discuss the relation with these earlier works. 2) The simulations are done on finite systems; how does momentum appear as a good quantum number? Did the authors use periodic boundary conditions? 3) In Sec. 4 it would be good to give some description of the system under investigation; could the Hamiltonian H_0 be defined? From the caption of Fig. 4 I gather that this is a Hubbard chain (at half filling?), but this should be discussed in the main text. 4) Also, do you prepare the system corresponding to H_0 in the ground state at t=0? This should be specified somewhere. 5) The authors introduce a projector to enforce the presence of a single source photon. How is this done in practice, i.e. in the DMRG setup? 6) It is well-known that MPS-based methods are limited in their simulation of time evolution, because of the growth of entanglement in the state. How is this limitation at work in the present setup. Are there other limitations? It would be good if the authors discuss the limitations of the method, such that the reader can judge the applicability for more challenging models such as 2D-cylinder Hubbard models. Is the performance in the end similar to "standard" MPS time evolution methods for e.g. ARPES spectral functions?

  • validity: good
  • significance: good
  • originality: high
  • clarity: low
  • formatting: good
  • grammar: good

Author:  Adrian Feiguin  on 2023-03-31  [id 3530]

(in reply to Report 1 on 2023-03-17)

We thank the Referee for carefully reading the manuscript and insightful questions about the method. We proceed to address them one by one, hoping that the improved version can be recommended for publication:

1) Is the method, in fact, a straightforward continuation of these earlier works? It would be good to discuss the relation with these earlier works.

Answer: This work precedes Ref. 44, as can be seen in the arXiv submission. However, for one those mysteries of destiny, it was published earlier. Ref. 44 was inspired by the ideas presented in this manuscript. Besides some conceptual connections (the idea of simulating spectroscopies in time, instead of doing the calculation in frequency), X-ray spectroscopies and notoriously more complex and require to account for additional degrees of freedom (core orbitals). Leaving this details aside, we have added a sentence in the introduction: "The idea of simulating the problem in real time instead of carrying out notoriously difficult calculations in the frequency domain have been tested in the simpler context of photoemission\cite{Zawadzki2019} and neutron scattering\cite{Zawadzki2020}. We hereby generalize them to the much more complex case of X-ray spectroscopies."

2) The simulations are done on finite systems; how does momentum appear as a good quantum number? Did the authors use periodic boundary conditions?

Answer: We have added a clarifying sentence: After obtaining the ground state we connect the extended probe at time $t=0$ and measure the momentum distribution function $n_{b,d}(k,t)$ at the detector as a function of time. Since in our tDMRG simulations we use open boundary condition, the proper definition of momenta corresponds to particle in a box states $\sin{(k_jx)}$ with momenta $k_j=j\pi/(L+1)$ with $ (j=1,\cdots,L)$. However, as customary in DMRG calculations, we vary $k$ continuously."

3) In Sec. 4 it would be good to give some description of the system under investigation; could the Hamiltonian H_0 be defined? From the caption of Fig. 4 I gather that this is a Hubbard chain (at half filling?), but this should be discussed in the main text.

Answer: We are very grateful to the Referee for bringing this omission to our attention! We have added an entire description or the model with additional simulation details directly under "Results".

4) Also, do you prepare the system corresponding to H_0 in the ground state at t=0? This should be specified somewhere.

Answer: This was answer in 2) above. We also clarify that this formulation does not depend on the system being in the ground state. In fact, the actual most outstanding feature of the method is that it also works away from equilibrium.

5) The authors introduce a projector to enforce the presence of a single source photon. How is this done in practice, i.e. in the DMRG setup?

Answer: We have added the explicit form of the projector in the sentence:

"This can be done by means of a projector: \begin{eqnarray} H_{source}= -|\kin\rangle \langle \kin|+\lambda \sum_{ij}n_{b,s,i}n_{b,s,j}, \end{eqnarray} where $|\kin\rangle \langle \kin|=n_{b,s}(\kk)=\frac{1}{L}\sum_{mn}e^{i\kin(\mathbf{R}m-\mathbf{R}_n)}b^\dagger$}b_{s,n"

"The full calculation proceeds as follows: The system is first initialized in the ground state of $H_0+H_{source}$...."

6) It is well-known that MPS-based methods are limited in their simulation of time evolution, because of the growth of entanglement in the state. How is this limitation at work in the present setup. Are there other limitations? It would be good if the authors discuss the limitations of the method, such that the reader can judge the applicability for more challenging models such as 2D-cylinder Hubbard models. Is the performance in the end similar to "standard" MPS time evolution methods for e.g. ARPES spectral functions?

Answer: While we do not show the evolution of the entanglement as a function of time, we have kept it under control by using a sufficiently large number of states (bond dimension). We have added the following discussion in the summary:

"In our calculations accuracy is kept it under control by using a sufficiently large number of DMRG states (bond dimension). Notice that at time $t=0$ the system in in the ground state of $H_0+H_{source}$. The core orbitals are in a product state of double occupied states and do not contribute to the entanglement. As time evolves, one electron will be excited from the core-orbitals, and one photon will eventually be emitted when the core hole recombines. When this occurs, the core orbitals return to a product state. Moreover, there is no hopping for the core degrees of freedom. This means that any additional entanglement will stem from the perturbations left behind in the system (which is a gapped Mott insulator) and the single photon at the detector, which will contribute to the entanglement by a bounded amount $\mathcal{O}(1)$. As a consequence, the entanglement growth will be minimal and simulations can proceed to quite long times. While we have not done a detailed quantitative analysis, we believe that the entanglement growth will be comparable, if not lower, than typical tDMRG simulations of spectral functions, particularly in the case of single-site RIXS."

---

## Round 2 · Referee Report · Anonymous (Referee 2) · 2023-9-2

Report

I would like to thank the authors for this revised version, I believe the manuscript is ready for publication.

---

## Round 2 · Author Response

We thank the Referees for their insightful and rigorous assessment of our manuscript. We greatly value their attention to detail that has helped us to greatly improve the manuscript. We are also pleased that both consider it worthy of publication.

---

## Round 2 · List of Changes

-We have added a clarifying sentence: "After obtaining the ground state we connect the extended probe at time $t=0$ and measure the momentum distribution function $n_{b,d}(k,t)$ at the detector as a function of time. Since in our tDMRG simulations we use open boundary condition, the proper definition of momenta corresponds to particle in a box states $\sin{(k_jx)}$ with momenta $k_j=j\pi/(L+1)$ with $ (j=1,\cdots,L)$. However, as customary in DMRG calculations, we vary $k$ continuously."

-We have added an entire description or the Hubbard model with additional simulation details directly under "Results".

-We have added the explicit form of the projector in the sentence:

"This can be done by means of a projector: \begin{eqnarray} H_{source}= -|\kin\rangle \langle \kin|+\lambda \sum_{ij}n_{b,s,i}n_{b,s,j}, \end{eqnarray} where $|\kin\rangle \langle \kin|=n_{b,s}(\kk)=\frac{1}{L}\sum_{mn}e^{i\kin(\mathbf{R}m-\mathbf{R}_n)}b^\dagger$}b_{s,n"

"The full calculation proceeds as follows: The system is first initialized in the ground state of $H_0+H_{source}$...."

-We have added the following discussion in the summary: "In our calculations accuracy is kept it under control by using a sufficiently large number of DMRG states (bond dimension). Notice that at time $t=0$ the system in in the ground state of $H_0+H_{source}$. The core orbitals are in a product state of double occupied states and do not contribute to the entanglement. As time evolves, one electron will be excited from the core-orbitals, and one photon will eventually be emitted when the core hole recombines. When this occurs, the core orbitals return to a product state. Moreover, there is no hopping for the core degrees of freedom. This means that any additional entanglement will stem from the perturbations left behind in the system (which is a gapped Mott insulator) and the single photon at the detector, which will contribute to the entanglement by a bounded amount $\mathcal{O}(1)$. As a consequence, the entanglement growth will be minimal and simulations can proceed to quite long times. While we have not done a detailed quantitative analysis, we believe that the entanglement growth will be comparable, if not lower, than typical tDMRG simulations of spectral functions, particularly in the case of single-site RIXS."

  • We have corrected the notation for Eq.(8)

-We have fixed the notation: "the system absorbs a photon with energy $\win$ and momentumm $\kin$ and emits another one with energy $\wout$, momentum $\kout$. We hereby focus on the so-called ``direct RIXS'' processes, see Fig.\ref{fig:fig2} and Fig.~1 in Ref.\cite{Kourtis2012-PhysRevB.85.064423}). As a consequence, the photon loses energy $\Delta \omega=|\wout-\win|=\win-\wout$ (from now one referred-to as simply $\omega$) and the electrons in the solid end up in an excited state with momentum $\kout-\kin$. In the following, we consider $\kin=0$ and refer to the momentum transferred simply as $\kk$."

  • Modified the sentence: "By an appropriate choice of $\Gamma_s^{\sigma\sigma'}=\Gamma_d^{\tau\tau'}=\Gamma$, and all others set to zero, one evolves the system in time to obtain a wave-function $|\psi_{\sigma\sigma',\tau\tau'}(t)\rangle$. This allows us to resolve the different contributions to the spectrum that split into spin conserving and non-conserving ones:..."

  • Modified legends in Fig. 4 and Fig. 5

  • We have expanded our description of Fig.5 in the text.

---

## Editorial Decision

published